

# Estimation of 1 km Grid-based WATEM/SEDEM Sediment Transport Capacity Using 1 Minute Rainfall Data and SWAT Semi-distributed Sediment Transport Capacity Results for Han River Basin of South Korea

**Chung-Gil Jung[1], Won-Jin Jang[2], Seong-Joon Kim[3,*]**

[1]Dept. of Civil, Environmental System Eng., Konkuk University, Seoul, 143-701, South Korea; wjd0823@konkuk.ac.kr
[2]Dept. of Civil, Environmental and Plant Eng., Konkuk University, Seoul, 143-701, South Korea; jangwj@nate.com
[3,*]Dept. of Civil, Environmental and Plant Eng., Konkuk University, Seoul, 143-701, South Korea; kimsj@konkuk.ac.kr

*Correspondence to*: kimsj@konkuk.ac.kr; Tel: +82-02-444-0186

**Abstract.** When assessing the total sediment yield of a watershed through sediment transport from soil erosion process, the ratio of sediment delivery is a critical and uncertain factor during modelling. This study is to estimate watershed scale sediment yield distribution of 1 km by 1 km spatial resolution with the evaluation of RUSLE (Revised Universal Soil Loss Equation) rain erosivity (R factor) for 14 years (2000 ~ 2013) using 1 minute data from 16 rainfall gauging stations in Han River basin (34,148 km$^2$) of South Korea. The WATEM/SEDEM sediment delivery algorithm based on RUSLE R, soil erodibility K,

length-slope LS factors was adopted. The average R factor values of 1 minute for the basin were evaluated as 3,812 MJ/ha mm/year. To determine the 1 km grid-based KTC (transport capacity coefficient generally given as 100) for the WATEM/SEDEM sediment transport estimation, the SWAT (Soil Water Assessment Tool) MUSLE (Modified USLE) results from 181 sub-watersheds (from 50 km$^2$ to 300 km$^2$) were used. The SWAT simulated suspended solids versus observed ones at 7 locations showed average R$^2$ (determination of coefficient) of 0.72. Using the SWAT sediment yields, the spatial KTC

based on 60 minutes R factor was determined at each sub-watershed from 0.16 to 142.6 with average value of 12.7 for the whole basin.

## 1 Introduction

Soil erosion and sedimentation by water involves the processes of detachment, transportation, and deposition of sediment by raindrop impact and flowing water (Foster and Meyer, 1977; Wischmeier and Smith, 1978; Julien, 1998). The major forces

originate from raindrop impact and flowing water. The mechanisms of soil erosion, in which water from sheet flow areas runs together under certain conditions and forms small rills. The rills make small channels. When the flow is concentrated, it can cause some erosion and much material can be transported within these small channels. A few soils are very susceptible to rill erosion. Rills gradually join together to form progressively larger channels, with the flow eventually proceeding to some established streambed. Some of this flow becomes great enough to create gullies. Soil erosion may be unnoticed on exposed





soil surfaces even though raindrops are eroding large quantities of sediment, but erosion can be dramatic where concentrated flow creates extensive rill and gully systems (Kim, 2006).

Extremely heavy rainfall events have increased over the past few decades (IPCC 2007) and been supported by observations in South Korea due to climate change. This climate change will result in changes in land ground cover type, biomass, and

hydrologic regimes and subsequently affect erosion on hillslopes. Topsoil removal reduces the productivity of land, while sediment-bounded nutrients increase the growth and proliferation of aquatic organisms such as algae (off-farm impact) (Pionke and Blanchard 1975); the suspended sediment then deteriorates the function of hydraulic structures. From this perspective, an accurate estimation of eroded soil is essential for hydrology, hydraulics, agriculture, and the ecosystem (Lee and Lee, 2010).

The soil erosion and sedimentation require a basic understanding of the spatial patterns, rates and processes of soil erosion

and sediment transport at the watershed scale. However, spatial data is often scarce possibilities to model spatial patterns of sediment delivery and to identify source areas of sediment are very limited (Haregeweyn et al., 2013). When a precipitation event occurs, the eroded soil is transported by a number of routes into local streams (Maidment, 1999). The soil erosion is related to the water flow that eventually reaches the saturated overland flow, which is controlled by the abundance and type of vegetation and underlying soil. Therefore, only some of the eroded soil is routed to the basin outlet. The ratio between the

basin sediment yield at the basin outlet and soil erosion over the basin is called the sediment delivery ratio. The sediment delivery ratio needs to be determined to generate the sediment. However, it can't be easily measured.

To overcome these problems, spatially distributed, process based models can be used. Several attempts have been made to use such process-based models in Ethiopia such as the Water Erosion Prediction Project model (Gete, 1999; Haregeweyn et al., 2013), the Agricultural Nonpoint Source Pollution model (Haregeweyn and Yohannes, 2003; Hussen et al., 2004) or the

Limburg Soil Erosion Model (Hengsdijk et al., 2005). However, such models require large amounts of input data whereas the return in increased accuracy of soil erosion prediction is limited (Jetten et al., 2003). If such models are applied in conditions where the necessary data are not available and/or a proper calibration cannot be performed, the results may become completely unreliable (Nyssen et al., 2006; Haregeweyn et al., 2013). Spatially distributed empirical or conceptual models may form an alternative to the complex physics-based spatially distributed models. WATEM/SEDEM (WAter and Tillage Erosion

Model/SEdiment DElivery Model) was developed for prediction of sediment yield at the catchment scale with limited data requirements (Van Oost et al., 2000; Van Rompaey et al., 2001). WATEM/SEDEM has been used in various types of environments in (Van Rompaey et al., 2001, 2003, 2005; Verstraeten et al., 2002, 2007), including hydrological catchments in Spain (de Vente et al., 2008; Alatorre et al., 2010). The Soil and Water Assessment Tool (SWAT) has been employed widely to evaluate the impact on soil erosion and sediment flux (Zhu et al., 2008). For example, Li et al. (2011) applied SWAT to

evaluate the effect of temperature change on water discharge, and sediment and nutrient loading in the Lower Pearl River basin, China. Hanratty and Stefan (1998) and Boorman (2003) have also described the application of SWAT to evaluate the impact of climate change on sediments in an agricultural watershed in Minnesota and in five European catchments.

The objective of this study is to estimate KTC (Transport Capacity Coefficient) of TC equation in WATEM/SEDEM algorithm with the evaluation of RUSLE (Revised Universal Soil Loss Equation) rain erosivity (R factor) for 15 years (2000





~ 2013) using 1 minute data from 16 rainfall gauging stations in Han River basin (34,148 km[2]) of South Korea. The KTC is traced by the sediment delivery of SWAT model determined by comparing MUSLE (Modified USLE) based SWAT (Soil Water Assessment Tool) simulated sediment yield.

## 2 Materials and methods

### 2.1 Sediment delivery model

We use TC equation in WATEM/SEDEM algorithm to estimate soil erosion and sediment flux to the stream network. The TC equation is a sediment delivery model based on the RUSLE model that predicts how much sediment is transported to the river channel on an annual basis. It is a spatially semi-distributed model, which means that the landscape is divided into small spatial units or grid cells (Van Oost et al., 2000; Van Rompaey et al., 2001; Verstraeten et al., 2002). The TC equation given by the

following equations:

$$TC = KTC \cdot R \cdot K \cdot (LS - 4.1s^{0.8}),  \tag{1}$$

where TC is the transport capacity (kg/m$^2$/yr) that means sediment delivery, R is the rainfall erosivity factor (MJ·mm/m$^2$·h·yr), LS a slope-length factor (Desmet and Govers, 1996), $s$ is the slope gradient, and KTC is an empirical transport capacity coefficient that depends on the soil and geomorphic characteristics in watershed. For distribution of sediment delivery, the

original algorithm has to be modified to fully distributed model.

The mass balance approach of the distributed model is followed for determining the net amount of sediment in each cell. The sediment transported to the cell from neighboring upslope cells is added to the sediment generated in cell by erosion, and this amount is then exported entirely to the downslope cells.

### 2.2 SWAT model

SWAT is a physically based, continuous, long-term, distributed-parameter model designed to predict the effects of land management practices on hydrology and water quality in agricultural watersheds under varying soil, land use, and management conditions (Arnold et al., 1998). SWAT is based on the concept of hydrologic response units (HRUs), which are portions of a sub-basin with unique land use, management, and soil attributes. The runoff, sediment, and nutrient loadings from each HRU are calculated separately based on weather, soil properties, topography, vegetation, and land management and are then summed

to determine the total loading from the sub-basin (Neitsch et al., 2002; Park et al., 2011; Park et al., 2014). The hydrologic cycle, as simulated by SWAT, is based on the water balance equation:

$$SW_t = SW_0 + \sum_{i=1}^{t}(R_{day} - Q_{surf} - E_a - W_{seep} - Q_{gw}),  \tag{2}$$

where $SW_t$ is the final soil water content (mm), $SW_0$ is the initial soil water content on day i (mm), t is the time (days), $R_{day}$ is the amount of precipitation on day i (mm), $Q_{surf}$ is the amount of surface runoff on day i (mm), $E_a$ is the amount of





evapotranspiration on day i (mm), $W_{seep}$ is the amount of water entering the vadose zone from the soil profile on day i (mm), and $Q_{gw}$ is the amount of return flow on day i (mm).

SWAT model estimate erosion and sediment yield for each HRU with the MUSLE (Modified Universal Soil Loss Equation). This includes the particle detachment by rainfall and flow, transport of particles by flow and deposition based on flow power and particle size that effect the ability of flow to continue to pick up particles or depositing them. The MUSLE (Williams, 1975; Neitsch et al. 2010) is considered in SWAT to estimates the erosion produced by both rainfall and surface runoff flow for each single rain storm in the following equation:

$$Sed = 11.8 \times \left( Q_{surf} \times q_{peak} \times area_{hru} \right)^{0.56} \times K_{usle} \times C_{usle} \times P_{usle} \times LS_{usle} \times CRFG , \tag{3}$$

where $Q_{surf}$ is the surface runoff volume (mm/ha), $q_{peak}$ is peak runoff rate (m3/s), $area_{hru}$ is the hydrologic response unit area (ha), $K_{usle}$ is the soil erodibility factor of the USLE, $C_{usle}$ is the cover management factor of USLE, $P_{usle}$ is the USLE support practice factor, $LS_{usle}$ is the USLE topographic factor, and $CRFG$ is the coarse fragment.

### 2.3 Study area description

The Han River Basin (34,148 km²) is one of the five major river basins in South Korea (99,720 km²). It occupies approximately 31 % of the country and falls within the latitude-longitude range of 36.03° N to 38.55° N and 126.24° E to 129.02° E. The basin has two main rivers, the North Han River (11,342 km²) and the South Han River (12,577 km²), which then merge and flow into the metropolitan city of Seoul, which has 10 million residents. The water resources of the river basin must be managed sustainably due to the expanding water demand of and supply to the Seoul area, including its satellite cities (12 million individuals), and the potential changes to water resources due to climate change must be evaluated (Ahn and Kim, 2015). Over the 30 years of weather data from 1985 to 2014, the average annual precipitation is 1,254 mm, and the annual mean temperature is 11.5 °C. The SWAT model is calibrated and validated for total Han River basin, however, estimation of KTC is evaluated by sediment delivery in focus area because suspended solid stations locate in South Han River (Figure 1).

In this study, four multipurpose dams (Hoengseong, Soyang, Chungju and Paldang) and three multifunction weirs (Kangcheon, Yeoju and Ipo) are selected as model calibration points (Figure 1). The Paldang dam is managed by KHNP (Korea Hydro & Nuclear Power Co., Ltd.), and other dams were managed by K-water (Korea Water Resources Corporation). The Hoengseong dam (HSD) and Chungju dam (CJD), located in the upstream region of the South Han River Basin, have sub-basin areas of 209 km² and 6,662 km² and storage capacities of 87 million m³ and 2.8 billion m³, respectively. Its storage capacity makes CJD the second largest dam in South Korea. The Soyang dam (SYD), located upstream in the North Han River Basin, has a storage capacity of 2.9 billion m³, making it the largest in South Korea, and a contributing sub-basin area of 2,694 km². The Kangcheon weir (KCW), Yeoju weir (YJW) and Ipo weir (IPW) were constructed by the government in 2012 to secure water resources and prevent flooding. These weirs are directly linked to the Paldang dam (PDD), which can supply more than 2.6 million m³ of water per day to Seoul and its metropolitan areas, with a storage capacity of 244 million m³.





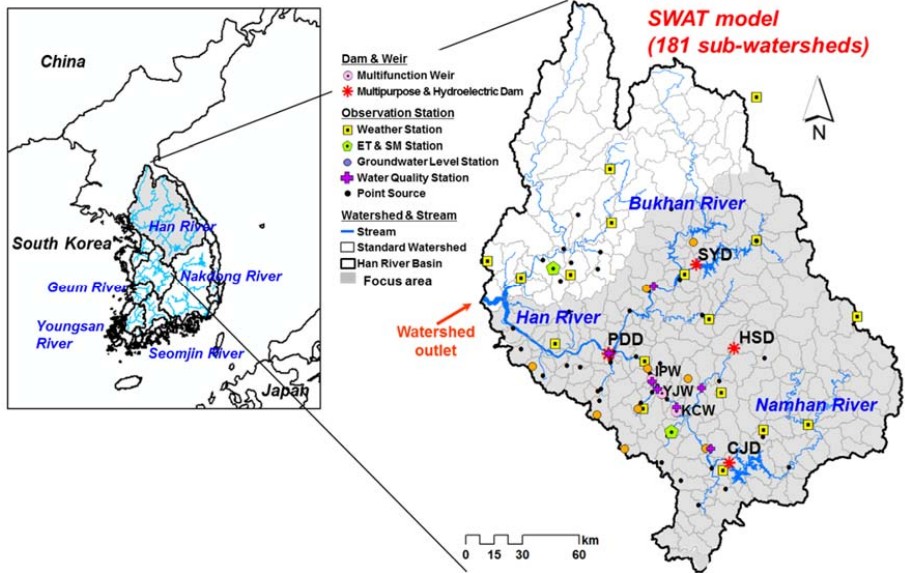

**Figure 1: Locations of the Han River basin (34,148 km²) and gauging stations and focus area for the hydrological modeling.**

### 2.4 Model implementation

To estimate spatially sediment delivery, KTC of TC equation has to apply observed empirical coefficient. However, it is very

difficult to directly observe KTC throughout widely area. In general, KTC only applied one value (100) acquired from specific experimental site. This study need observed KTC for accurately estimating sediment delivery of TC equation. SWAT is a physically based, continuous, long-term, distributed-parameter model designed to predict the effects of land management practices.

For accurately estimating observed KTC, this study use SWAT model results. SWAT model verify simulated suspended

solid (SS) from observed SS at seven water quality stations in river. From the SWAT results, each sediment delivery of sub-watershed assumes that the sediment delivery is corrected by calibrating SS in river.

The study procedure is that the distributed sediment delivery firstly is calculated by TC equation. Secondly SWAT model verify simulated SS from observed SS at seven water quality stations in river. Finally, KTC are determined by comparing MUSLE based SWAT (Soil Water Assessment Tool) simulated sediment yield. Using the SWAT sediment yields from 181

sub-watersheds of the basin, the KTCs of sediment delivery model are determined at each sub-watershed (Figure 2).





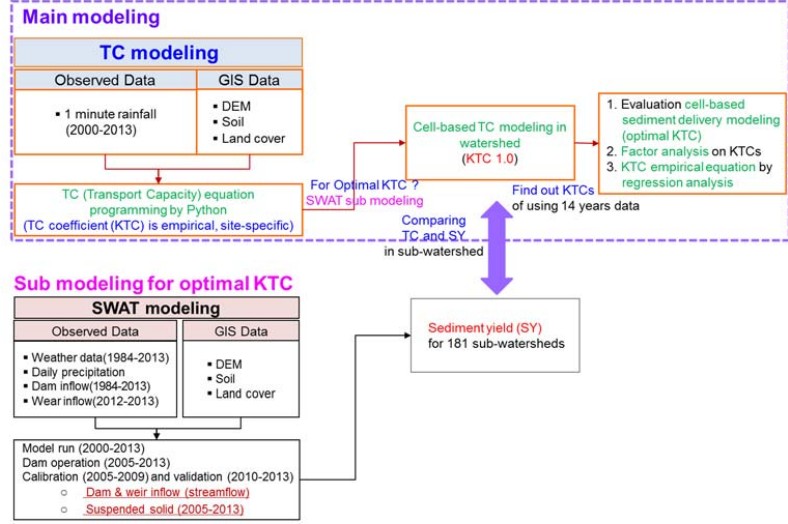

**Figure 2: The study procedure.**

## 3 Results and discussion

### 3.1 Sediment delivery model for evaluation

5    TC equation in The yearly distributed R factor was evaluated by 1 minute rainfall data with 1 km ×1 km grid cell. The model estimated an average R factor of 3,812 MJ/ha mm/year during 2000-2013 in the overall focus area (Figure 3). Also, the range of maximum R factor during 2000-2013 was 1438.6 ~ 13616.1 MJ/ha mm/year.

The yearly distributed sediment delivery (kg/m²/year) based on TC equation was evaluated by KTC value (1.0) as initial condition with 1 km ×1 km grid cell corresponding to the sub-watershed scale. The model predicted an average sediment

10   delivery (SD) of 0.134 kg/m²/year during 2000-2013 in the overall focus area (Figure 4). Also, the range of maximum SD during 2000-2013 was 0.954 ~ 2.711 kg/m²/year. The regions seen from high sediment delivery can be explained by K factor, high LS factor and lots of highland agricultural land than other regions.




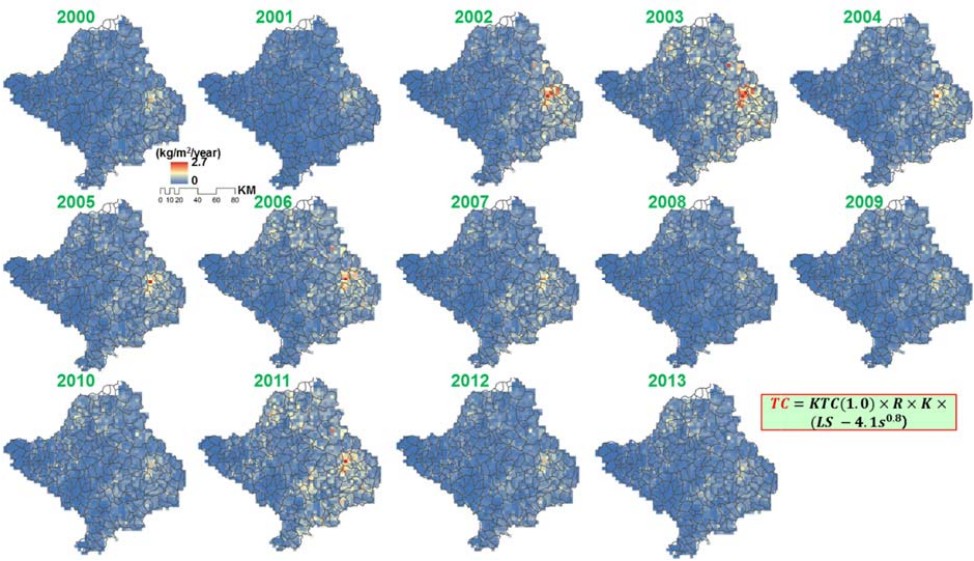

**Figure 3: The distribution of rain erosivity factor by 1 minute rainfall data during 2000-2013 year.**

5   **Figure 4: The distribution of sediment delivery predicted by TC modelling (KTC = 1.0) during 2000-2013 year.**



### 3.2 SWAT model for evaluation

The SWAT model was calibrated at seven locations (HSD, SYD, CJD, KCW, YJW, IPW, and PDD) in the main river reaches using five years (2005–2009) of daily inflow (streamflow) data to the dams and weirs and subsequently validated using another four years (2010–2013) of data using the average calibrated parameters (Figure 5). As seen the Figure 6, the model was

5 spatially calibrated and validated using evapotranspiration and soil moisture data measured at two locations over five years (2009–2013). In the case of dam inflow, the $R^2$ value was greater than 0.59. The average NSE was 0.59 at HSD, 0.78 at SYD, 0.61 at CJD, 0.79 at KCW, 0.77 at YJW, 0.88 at IPW, and 0.87 at PDD. The PBIAS of HSD, CJD, SYD, KCW, YJW, IPW and PDD were 13.5 %, 12.2 %, 9.4 %, 11.5 %, 19.8 %, 21.4 %, and 4.5 %, respectively. In the case of the dam storage volume, the average $R^2$ was between 0.40 and 0.96, and the PBIAS was between 0.9 % and 18.9% for each calibration point. The

10 average $R^2$ for evapotranspiration was between 0.77 and 0.72, and the soil moisture was between 0.80 and 0.78, and the groundwater level was between 0.47 and 0.68 for each calibration point (Table 1 and Table 2). Detailed results could refer to the Ahn and Kim (2016, under review). The calibration results checked with SWAT calibration guidelines (NSE ≥0.5, PBIAS ≤28%, and $R^2$ ≥0.6, Moriasi et al., 2007) and were found to be satisfactory.

In this study, the SWAT model was used to simulate SS (tons) within the Han River basin. The SWAT model was calibrated

at the same locations of streamflow in the main river reaches using five years (2005–2009) of eight days intervals SS data and subsequently validated using another five years (2010–2014) of data using the average calibrated parameters (Figure 7). The average $R^2$ for SS was between 0.61 and 0.80 for each calibration point. The average $R^2$ of the streamflow was typically greater than 0.60, which indicates a satisfactory simulation.




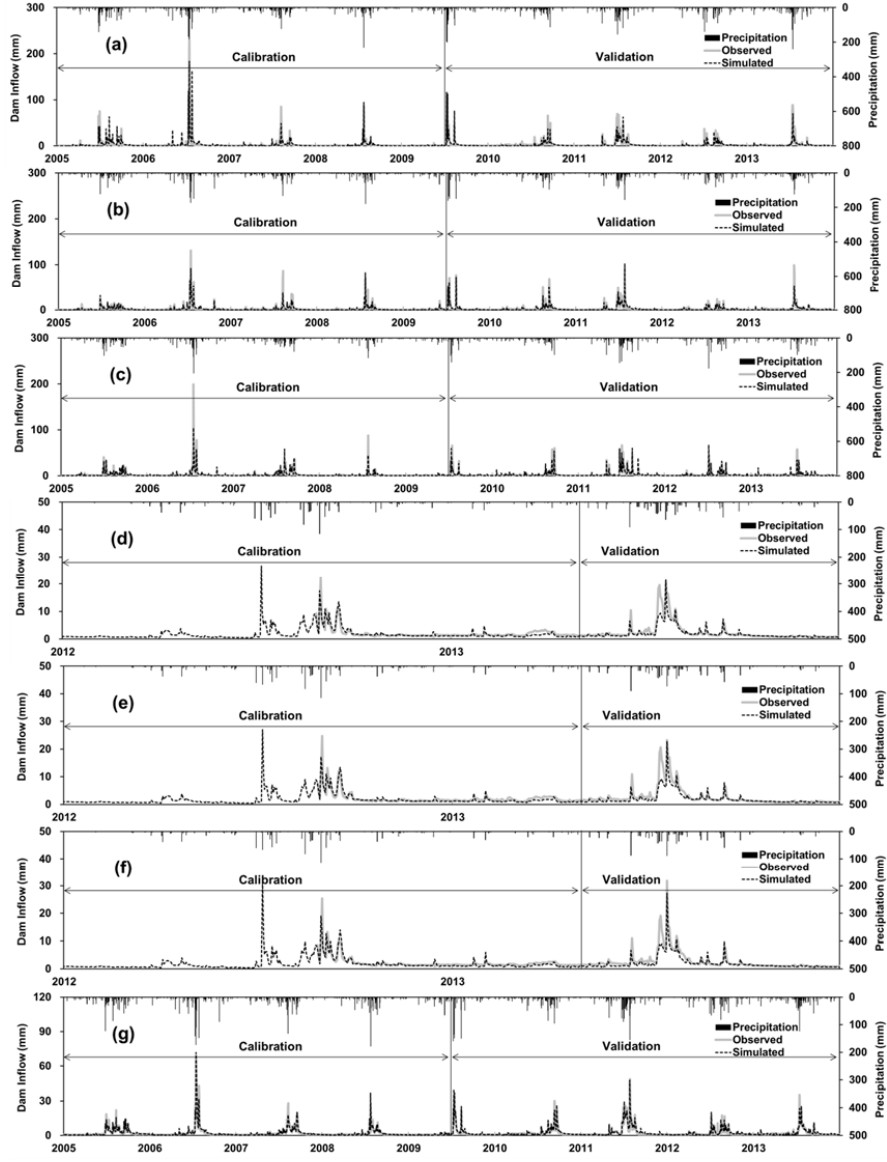

**Figure 5: Comparison of the observed and SWAT-simulated daily dam inflow (streamflow) during the calibration (2005–2009) and validation (2010–2013) periods at (a) HSD, (b) SYD, (c) CJD, (d) KCW, (e) YJW, (f) IPW, and (g) PDD.**





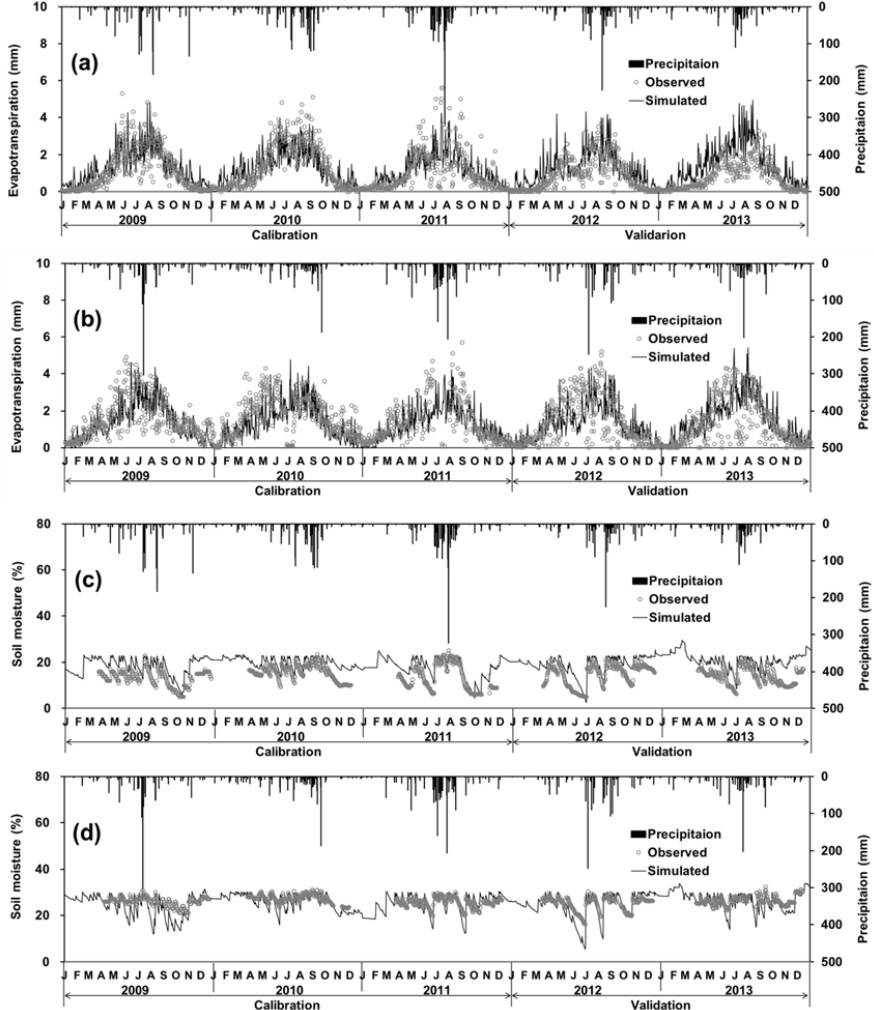

**Figure 6: Comparison of the observed and SWAT-simulated daily evapotranspiration at (a) SM and (b) CM and soil moisture at (c) SM and (d) CM during 2009–2013.**





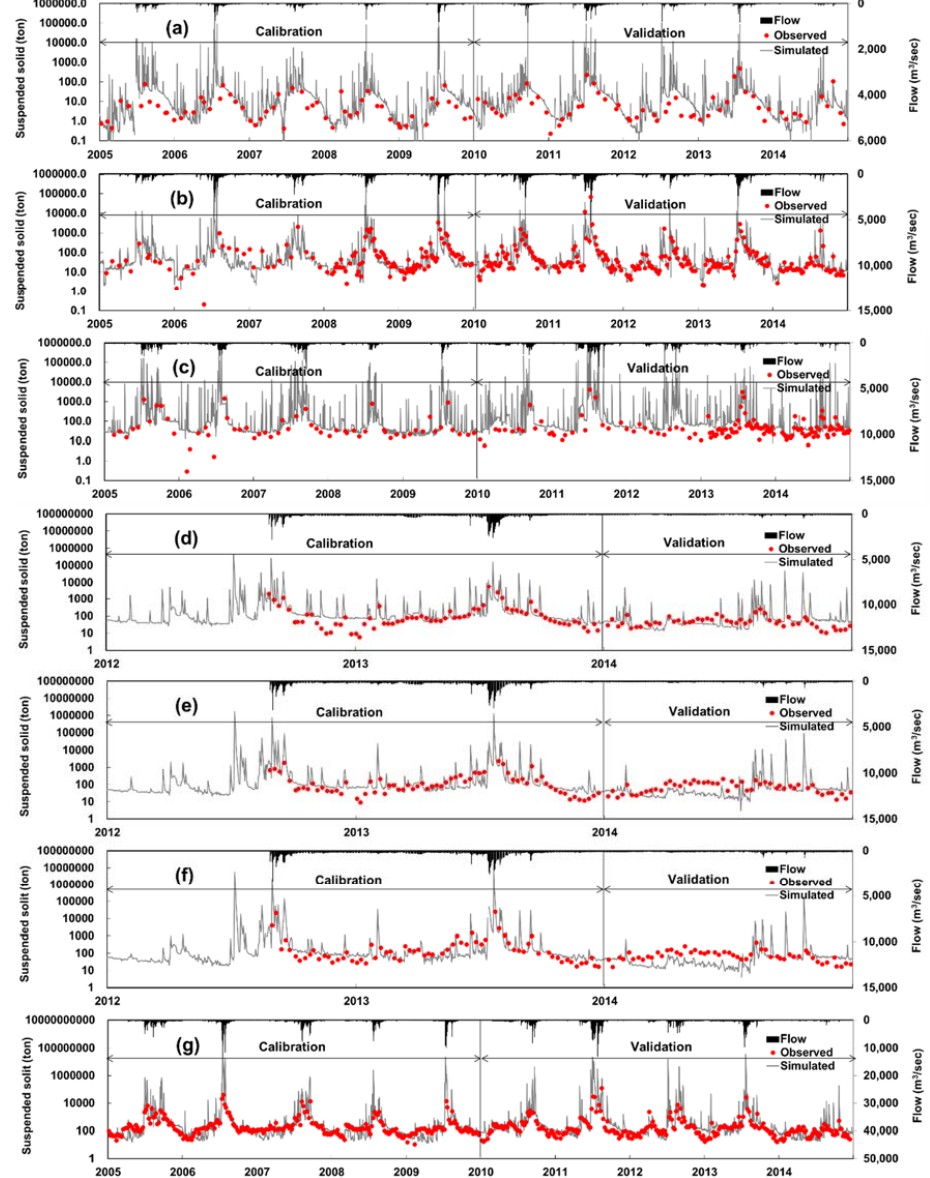

**Figure 7: Comparison of the observed and SWAT-simulated daily suspended solid during the calibration (2005–2009) and validation (2010–2014) periods at (a) HSD, (b) SYD, (c) CJD, (d) KCW, (e) YJW, (f) IPW, and (c) PDD.**



**Table 1.** Calibration and validation results for dam inflow and storage at seven calibration points in the main reach.

| Model output | Evaluation criteria | HSD | | SYD | | CJD | | KCW | | YJW | | IPW | | PDD | |
|---|---|---|---|---|---|---|---|---|---|---|---|---|---|---|---|
| | | Cal. | Val. | Cal. | Val. | Cal. | Val. | Cal. | Val. | Cal. | Val. | Cal. | Val. | Cal. | Val. |
| Dam inflow (mm) | $R^2$ | 0.82 | 0.84 | 0.90 | 0.89 | 0.81 | 0.74 | 0.90 | 0.63 | 0.91 | 0.62 | 0.93 | 0.59 | 0.92 | 0.88 |
| | NSE | 0.61 | 0.57 | 0.78 | 0.78 | 0.63 | 0.58 | 0.78 | 0.79 | 0.77 | 0.76 | 0.81 | 0.95 | 0.83 | 0.76 |
| | RMSE (mm/day) | 7.9 | 9.3 | 3.8 | 3.9 | 3.5 | 3.1 | 6.5 | 0.7 | 9.1 | 2.4 | 9.2 | 2.9 | 0.8 | 2.3 |
| | PBIAS (%) | 14.5 | 12.5 | 10.3 | 14.0 | 8.9 | 9.9 | 18.0 | 4.9 | 25.5 | 14.1 | 25.6 | 17.2 | 2.2 | 6.8 |

Cal. = calibration period (HSD, SYD, CJD and PDD: 2005-2009, KCW, YJW and IPW: 2013) and Val. = validation period (HSD, SYD, CJD and PDD: 2010-2014, KCW, YJW and IPW: 2014)

5 **Table 2.** Calibration and validation results for evapotranspiration and soil moisture at two calibration points and groundwater level fluctuation at five calibration points in the watershed.

| Model output | Evaluation criteria | SM | | CM | | GPGP | | YPGG | | YPYD | | YIMP | | HCGD | |
|---|---|---|---|---|---|---|---|---|---|---|---|---|---|---|---|
| | | Cal. | Val. | Cal. | Val. | Cal. | Val. | Cal. | Val. | Cal. | Val. | Cal. | Val. | Cal. | Val. |
| Evapotranspiration (mm) | $R^2$ | 0.81 | 0.73 | 0.70 | 0.74 | - | - | - | - | - | - | - | - | - | - |
| | NSE | 0.64 | 0.45 | 0.50 | 0.55 | - | - | - | - | - | - | - | - | - | - |
| | RMSE (mm/day) | 2.3 | 9.1 | 4.0 | 3.0 | - | - | - | - | - | - | - | - | - | - |
| | PBIAS (%) | 9.6 | 30.2 | 11.6 | 23.7 | - | - | - | - | - | - | - | - | - | - |
| Soil moisture (%) | $R^2$ | 0.85 | 0.75 | 0.78 | 0.78 | - | - | - | - | - | - | - | - | - | - |
| Groundwater level (El.m) | $R^2$ | - | - | - | - | 0.70 | 0.63 | 0.64 | 0.45 | 0.70 | 0.41 | 0.53 | 0.40 | 0.69 | 0.67 |

Cal. = calibration period (2009-2011) and Val. = validation period (2012-2013)

### 3.3 The KTCs for evaluation

To perform optimal value for KTCs of whole 181 sub-watersheds for 14 years (2000-2013), KTCs were controlled between
10 sediment delivery which was estimated as 1.0 value of KTC and sediment yield which was estimated by SWAT model calibrated with observed suspended solid. The KTCs were SWAT results divided by TC modelling results to each sub-watershed. The average KTC value was 12.58 and the ranges of KTCs were 0.16 ~ 112.58 (Figure 8).





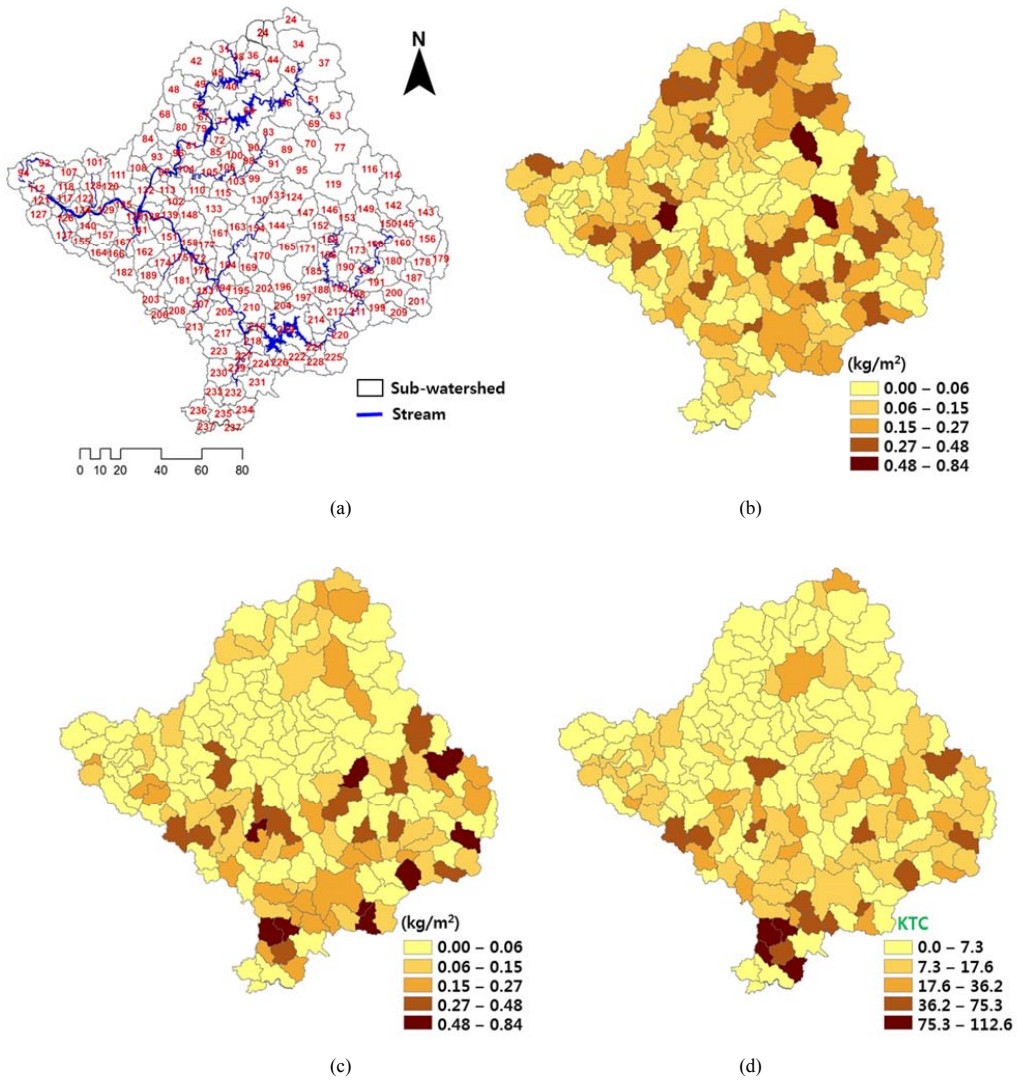

**Figure 8: The distribution map of KTC in each sub-watershed: (a) each sub-watershed number, (b) sediment delivery of TC modelling, (c) sediment delivery of SWAT modelling, and (d) estimated optimal KTCs.**




### 3.4 Analysis of impact factors for scenario1

The KTC varies with conditions corresponding to different land uses and watershed characteristics. In addition, it is very difficult to estimate optimal KTC to each sub-watershed. For estimation of optimal KTC, simple regression equation is demanded from impact factors.

The watershed slope, watershed area, and K factor were selected as impacted factors of KTC (Alatorre et al., 2012) and then analysed relationship of correlation between optimal KTC and impact factors. As the result of correlation analysis to area and K factor (scenario1), relationship of watershed slope and KTC showed very low Pearson's coefficient of 0.12. The watershed area and K factor showed high interrelationship with Pearson's coefficient -0.68 and 0.72 (Table 3). Therefore, watershed area and K factor could affect the KTC estimation (Figure 9). In this study, we divided that the sub-watershed area

was from 50 km$^2$ to 300 km$^2$. It is standard watershed scale in South Korea. If watershed area goes over this standard watershed scale, the result of regression analysis will differ from this study results.

As a result, 231, 233, 235, and 237 sub-watersheds show high KTC value as 92.2 that it has about 8.9 times than other KTCs. Because the KTCs showed low watershed area and high K factor value, KTC as well as sediment delivery showed high value. However, prediction results using regression analysis has high uncertainty in 231, 233, 235, and 237 watersheds (Figure

9(c)).

**Table 3.** Relationship between optimal KTC and influenced factors.

| Factor | Pearson's coefficient of correlation | Interrelationship |
| --- | --- | --- |
| Watershed slope | 0.12 | Very low |
| Watershed area | -0.68 | High |
| Watershed K factor | 0.72 | High |

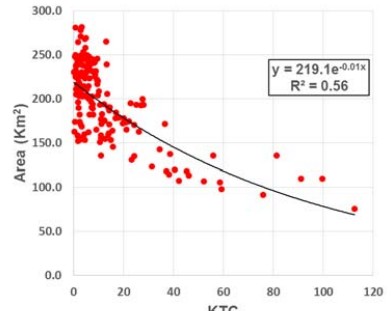
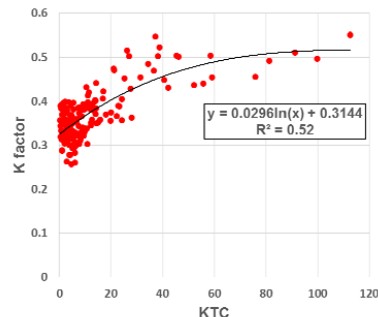





(a)                                                                                             (b)

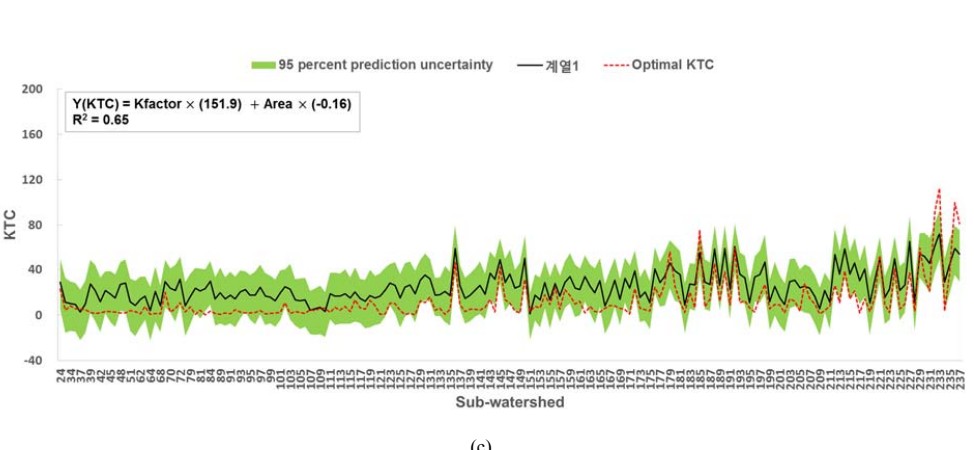

(c)

**Figure 9: The KTC regression analysis results for scenario1: (a) area and KTC (b) K factor and KTC, and (c) multiple regression analysis between area and K factor.**

### 3.5 Analysis of impact land use for scenario2

The results of TC modelling didn't fully reflect the effect of land use. As the equation apply K factor, the results can indirectly
considered effect of land use. In spite of the consideration, especially, the influence in regard to soil erosion on agricultural land has to be directly analysed.

In this study, we tried to find optimal KTC with K factor and watershed area by multiple regression analysis to each sub-watershed. Additionally, estimation of optimal KTC was analysed by considering ratio of land use to each sub-watershed (scenario2) (Figure 10(a) and 10(b)).

As the results of regression analysis for scenario2, relationship of agricultural land and KTC showed high relationship with $R^2$ of 0.51. However, relationship of rice paddy and KTC showed very low correlation. The regression equation considered ratio of agricultural land about optimal KTC differed the equation considered area and K factor (Fig. 10(c)). The $R^2$ for final optimal KTC by considering land use improve 0.11 as 0.76. From the results, the TC equation could be modified by optimizing KTCs considering ratio of agricultural land.

By comparing regression analysis considered area and K factor, the predicted KTC improved accuracy and reduced uncertainty. Also, uncertainty rate of KTC was reduced by 50 % at high KTC watershed against estimating the KTC regression analysis to consider watershed area and K factor. These results can be illustrated with statistical analysis (Table 4). The result of scenario2 was similar to statistical summary of optimal KTC.





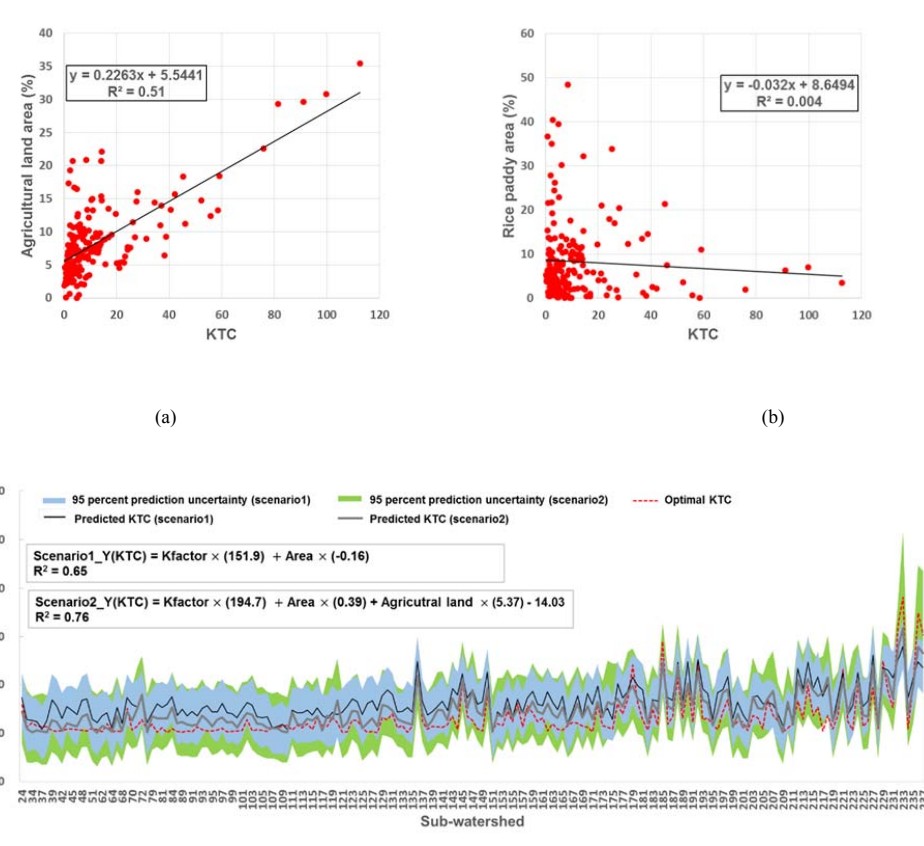

Figure 10: The KTC regression analysis results of scenario1 and scenario2: (a) Ratio of agricultural land and KTC (b) ratio of rice paddy and KTC, and (c) Multiple regression analysis between area, K factor, and ratio of agricultural land.

Table 4. Statistical summary for predicted and optimal KTC.

| Factor Scenario | Max. | Min. | SD | PE (%) |
|---|---|---|---|---|
| Scenario1 (area and K factor) | 72.1 | 1.2 | 14.0 | 50.2 |
| Scenario2 (area, K factor and agricultural land) | 87.5 | 0.08 | 15.3 | 32.8 |
| Optimal KTC (observed data) | 112.6 | 0.16 | 18.1 | |

Max.: maximum KTC, Min.: minimum KTC, SD: standard deviation, PE: percentage error


### 3.6 Programing and graphical user interface

The whole process in distribution modelling is needed to develop program in order to automatically estimate all the distributed results in this study. The developed program helped general users to easily apply estimated methods in this study (Figure 11). The rain erosivity program is divided into four step process. Firstly, rainfall data type select whether this study used 1 minute

5   or not 1 hour rainfall data. Secondly, this program automatically selects rainfall events and maximum rainfall duration. Thirdly, rain erosivity and regression analysis are calculated. Finally, the program spatially distributes the results in the study area. Also, the TC calculation program is divided into 3 step process. Input data is required with estimated rain erosivity (R factor) and slope map. The K factor and LS factor automatically are calculated by this tool. This program estimates KTC map by multiple regression equation. Finally, TC map is calculated by estimated KTC map.

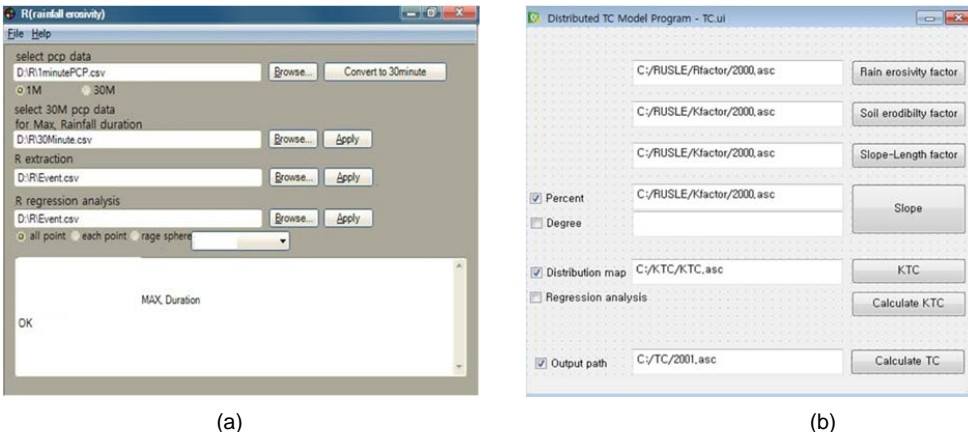

(a)                                         (b)

**Figure 11: The development of distributed modelling programs: (a) Rain erosivity modelling (b) TC calculation modelling.**

### 4 Summary and conclusion

15  This study attempted to estimate KTC (Transport Capacity Coefficient) of TC equation in WATEM/SEDEM algorithm with the evaluation of RUSLE (Revised Universal Soil Loss Equation) rain erosivity R factor for 15 years (2000 ~ 2013) using 1 minute data from 16 rainfall gauging stations in Han River basin (34,148 km$^2$) of South Korea. The KTC was traced by the sediment delivery of SWAT model determined by comparing MUSLE (Modified USLE) based SWAT (Soil Water Assessment Tool) simulated sediment yield. The SWAT model was calibrated and validated with average R$^2$ of 0.72 using 10 years

20  observed SS (suspended solid). Using the SWAT sediment yields from 181 sub-watersheds of the basin, the KTCs of sediment delivery model were determined at each sub-watershed.





In general, the KTC was used as a fixed value of 100 as an empirical constant. The KTCs were controlled between sediment delivery which was estimated as 1.0 value of KTC and sediment yield which was estimated by SWAT model calibrated with observed suspended solid. The KTC value averagely was 12.58 and the ranges of KTCs were 0.16 ~ 112.58. The KTC varies with conditions corresponding to different land uses and watershed characteristics. In addition, it is very difficult to estimate

optimal KTC to each sub-watershed. For estimation of optimal KTC, simple regression equation is demanded from impact factors. As the result of correlation analysis to area and K factor (scenario1), relationship of watershed slope and KTC showed very low Pearson's coefficient of 0.12. The watershed area and K factor showed high interrelationship with Pearson's coefficient -0.68 and 0.72. The results of TC modelling didn't fully reflect the effect of land use. As the results of regression analysis for scenario2, relationship of agricultural land and KTC showed high relationship with $R^2$ of 0.51. However,

relationship of rice paddy and KTC showed very low correlation. The regression equation considered ratio of agricultural land about optimal KTC differed the equation considered area and K factor. The $R^2$ for final optimal KTC by considering land use improve 0.11 as 0.76. From the results, the TC equation could be modified by optimizing KTCs considering ratio of agricultural land.

Based on the distributed results of the model, we could determine identification of KTC in watershed scale under various

topographic conditions. The area and K factor and ratio of agricultural land area were more strongly affected, and the KTC values were proportional to area, K factor, and agricultural land. Further studies are necessary to understand KTC changes from various conditions, for example, forestation density and the urban density affected directly K factor. Understanding the factors contributing to KTC progression can help in the development of mitigation strategies for future climate change.

*Acknowledgements*. This research was supported by a grant (16AWMP-B079625-03) from the Water Management Research Program funded by the Ministry of Land, Infrastructure and Transport of the Korean government.

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
