# Peer review of "Estimation of 1 km Grid-based WATEM/SEDEM Sediment Transport Capacity Using 1 Minute Rainfall Data and SWAT Semi-distributed Sediment Transport Capacity Results for Han River Basin of South Korea"

_Hydrology and Earth System Sciences, 2016_

## Referee Comment (RC1) · Anonymous Referee #1 · 3 Feb 2017

General comments:

In the submitted paper authors combined the WATEM/SEDEM and SWAT models. The main objective of this study was to estimate the transport capacity parameter that is used in the WATEM/SEDEM model to determine the maximum amount of sediments (annual transport capacity) that can be transported through each grid cell. The parameter was estimated based on the SWAT model results. As part of the study high-frequency rainfall data (1-minute time step) was used to determine the rainfall erosivity

in the investigated Han River Basin in South Korea. The SWAT model was calibrated using flow, evaporation, suspended sediment and soil moisture data.

The paper is in the scope of the journal and the presented topic could potentially be interesting for the society. However, there are several major drawbacks related to the submitted paper. Thus, I must suggest rejecting the paper in the current form due to the following reasons:

1) The paper structure is a bit confused, the main objective of the study or the scientific question is not clear enough. Further, what are actually the main conclusions of the study, what is the take home message of this paper? Moreover, the language should be significantly improved (grammar, overall style and structure because some sentences are not clear). In its current form the paper is not suitable for the Hydrology and Earth System Sciences journal.

2) The paper is a bit short and from my point of view there is nothing wrong if the paper is short, in case that all the steps are correctly explained. In the presented paper a lot of steps are not explained, for example: how was the rainfall erosivity determined based on the 1-minute rainfall data, what method was used to determine the spatial distribution of rainfall erosivity, how was the calibration of SWAT model carried out, more details about the data (sediment, soil moisture,...) used should be provided. Thus, I believe that the study is not reproducible.

3) The discussion of the results is poor and should be improved (e.g., what transport capacity parameters were used in previous WATEM/SEDEM model studies).

Specific comments and technical corrections:

Page 1, line 27: Which soils are susceptible to rill erosion? Please add more details.

Page 2, line 4: Some references should be added to confirm this statement.

Page 2, lines 4-5: I would suggest replacing "will" with "may", "might" or "could".

Page 2, line18: Why is only Ethiopia mentioned here? These kinds of models were also used in other countries.

Page 2, line 33: This sentence should be rephrased because the TC equation is defined on the next page of the manuscript.

Page 3, line 8: Which spatially semi-distributed model?

Page 3, line 9: Replace "TC equation given" with "TC equation is given".

Page 3, lines 14-15: This modification of the algorithm should be described.

Page 4, line 21: Replace "stations locate" with "stations are located".

Page 5, first paragraph of section 2.4: This paragraph should be rewritten because it is not clear.

Page 5, lines 9-10: How does the SWAT model verify measured suspended solids? Please rephrase this sentence.

Page 6, line 5: Replace "The yearly" with "the yearly".

Page 6, line 5: It should be described how was the rainfall erosivity calculated (which equations were used, how was the data measured, was 1-minute data really used or was the data aggregated,. . .).

Page 6, second paragraph of section 3.1: This part should also be rewritten because several things are not well explained, for example: what is meant by yearly distributed sediment delivery? When you are referring to sediment delivery, is this the sediment delivery ratio or something else because the sediment delivery ratio should not have any units? The TC parameter in the WATEM/SEDEM is the transport capacity parameter of each grid cell? How did you take into account the fact that the WATEM/SEDEM model should be used with grid resolution of 20 by 20 m and using different resolutions may cause problems (e.g., LS factor)? How was the LS factor calculated? How did you determine the soil erodibility factor?

Page 7, Fig. 3: How was spatially distributed rainfall erosivity calculated?

Page 7, Fig. 4: The TC parameter defines the maximum amount of sediments that can be transported through each grid cell.

Page 8, line 2: How was the calibration carried out?

Page 8, lines 6-7: NSE and PBIAS acronyms should be defined.

Page 8, lines 10-11: More information should be provided about the soil moisture data. It is stated that detailed results are available in the paper that is currently under review, but this paper is publically not available.

Page 8, line 15: What is meant by "eight days intervals"? Please explain.

Page 8, line 18: "indicates a satisfactory simulation". This is very subjective and I would suggest avoiding this kind of statements.

Page 9, Fig. 5: I would suggest using different colours for observed and simulated data (one could be red and the other one grey or black; or something similar).

Page 10, Fig. 6: I would suggest using different colours for observed and simulated data (one could be red and the other one grey or black; or something similar).

Page 11, Fig. 7: Are these really daily suspended solids because the density of points is a bit low (e.g., c) case). How were suspended solids measured and how reliable are measurements?

Page 12, first paragraph of section 3.3: This section should be better explained.

Page 14, lines 2-3: Why is it difficult to estimate the KTC and what is the uncertainty related to this estimation?

Page 14, lines 3-4: Please rephrase this sentence.

Page 14, lines 9-10: Please rephrase this sentence, because it is not clear.

Page 15, lines 15-16: This is subjective (what is high and what is low?).

Page 17, section 3.6: This section should be moved before the results and discussion part. From my points of view this should go into the methodology. More information about the software could be provided (e.g., is it publically available,...).

Page 17, line 17: "The KTC was traced" this is not clear, please rephrase.

Page 20, lines 7-8: This reference is not mentioned in text.

———————————————————

---

## Referee Comment (RC2) · Anonymous Referee #2 · 13 Feb 2017

The submission describes the estimation of transport capacity coefficient (KTC) in WATEM/SEDEM algorithm with the evaluation of RUSLE R factor using 1 min rainfall data in Han River basin of South Korea. The SWAT model, which includes the MUSLE function for calculating soil losses from the watershed, has been used to determine the WATEM/SEDEM sediment transport estimation. Studies such as this are relatively rare, and the model appears to be effectively calibrated and applied. This reviewer agrees that the manuscript contains novel information that could be useful for the readers of

[Figure]

HESS. Much of the theoretical development presented in this manuscript is clear and well described. However, it reads more like a book chapter than a journal article. It is because the authors present few theoretical background and discussion of results, implications and limitations. For example, there is a lack of information regarding how the variation of KTC could affect the sediment yield at the sub-watershed scale. Moreover, several sections of the manuscript are not connected well, and importantly it is hard to understand what the major findings are. Although I generally recommend the paper for publication in HESS, I have the following comments which have, to my opinion, to be considered in a revised version.

1. Introduction. P2.L29-32: it confuses me why you used such long content to introduce SWAT studies, which are not the key topic of your study. I would like see a clear hypothesis (framework) of your study, following introduction of your aim line P2.L33-P3.L3. Then, if essentially, introduce some method to test your hypothesis.

2. Study area description. P4.L13-21: please introduce rough annual distribution of precipitation and temperature, e.g. precipitation mostly occurred in some month, min and max temperature over year. Add a description of land use and soil data modeled in this study. How were point sources of sediment, N and P accounted for? Figure 1: Please remove the layers that were not used in model calibration.

3. Method: Authors should provide proper justification to consider this approach for possible use in other studies. The differences and limitations should be included in the Methodology.

4. Model implementation. P5.L4-15: More detail about soils how similar were the attributes (e.g. soil type) of the sub-watersheds. How were data for the individual KTC determined?

5. Results and discussion: Overall, the authors failed to provide a detailed report on the data obtained during the study and then need to discuss the importance of this study with regard to the relevant scientific or technical issues about sediment transport

capacity. In this section, the authors simply explained the outcomes from model simulation that could not support to the significant results. Discussion should be concise and add only essential points in terms of the current results and limitations.

6. Conclusions: The findings of this study will be more useful if the authors can address how these findings will impact the evaluation of sediment transport capacity. Conclusions could be better stated by a better interpretation of the data and model predictions.

---

## Author Comment (AC1) · 3 Apr 2017

**HESS-2016-649-R1**

We thank the reviewer for his constructive review and intend to address all of his comments. We would like to state that the presented paper included various study such as models, algorithm and regression analysis. In this paper, WATEM/SEDEM algorithm was firstly introduced to South Korea. Also, we think that KTC empirical equation would be useful at ungauged watershed. For resubmission of improved paper, we think that all the following comments went through from the previous reviews.

1) The paper structure is a bit confused, the main objective of the study or the scientific question is not clear enough. Further, what are actually the main conclusions of the study, what is the take home message of this paper? Moreover, the language should be significantly improved (grammar, overall style and structure because some sentences are not clear). In its current form the paper is not suitable for the Hydrology and Earth System Sciences journal.

- **Answer: We consent to your comments. For language problems, we plan to get the native English speaker review/proofread through American Journal Experts (payment is about $ 500). We will attach editing invoice.**

- **Answer: The paper is outlined as follows: Sect. 1 described application of WATEM/SEDEM algorithm in South Korea. However, KTC (Transport Capacity Coefficient) is necessary for application of WATEM/SEDEM algorithm in South Korea. So, Sect. 2 traced KTC by the sediment delivery of SWAT model determined as comparing MUSLE (Modified USLE) based SWAT (Soil Water Assessment Tool) simulated sediment yield. The SWAT model results reflected observed suspended solid. Sect. 3 find out KTC empirical equation by linear regression analysis. The KTC equation is going to be commonly used for accurate sediment delivery at a ungauged watershed in South Korea. Finally, calibrated spatial sediment delivery from WATEM/SEDEM algorithm is estimated using obtained KTCs by KTC empirical equation.**

- **The objective of this study is to estimate KTC empirical equation for calibrated spatial sediment delivery and to prove accuracy of sediment delivery by KTC empirical equation. Therefore, we consent to reviewer's comments. We are going to rewrite paper structure according to above purpose.**

2) The paper is a bit short and from my point of view there is nothing wrong if the paper is short, in case that all the steps are correctly explained. In the presented paper a lot of steps are not explained, for example: how was the rainfall erosivity determined based on the 1-minute rainfall data, what method was used to determine the spatial distribution of rainfall erosivity, how was the calibration of SWAT model carried out, more details about the data (sediment, soil moisture,...) used should be provided. Thus, I believe that the

study is not reproducible.

- **Answer: We consent to your comments. This paper was not explained about generation of 1 minute data and rainfall erosivity, calibration of SWAT model. Because methods for estimating rainfall erosivity and SWAT calibration are generally known in hydrology field, we don't mention detailed process. Also, we suggested a previous study thesis instead of detailed explanation, for example, the Ahn and Kim (2016). According to reviewer comment, we can give all the process in detail. Therefore, we will certainly explain generation of rainfall erosivity, data used in this study, and method of SWAT calibration in part of 2 Materials and methods.**
   **.**

3) The discussion of the results is poor and should be improved (e.g., what transport capacity parameters were used in previous WATEM/SEDEM model studies).

- **Answer: We consent to your comment. By reflecting 1 and 2 comments, discussion about results will be improved. Also, we will additionally analyze cause of error according to each sector. So, we will make up for the weak points in this paper.**

4) The discussion Specific comments and technical corrections: Page 1, line 27: Which soils are susceptible to rill erosion? Please add more details.

- **Answer: Thank you for your comment. We consent to your comment. We will certainly explain soil characteristics in regard to rill erosion factor.**

5) The discussion Specific comments and technical corrections: Page 2, line 4: Some references should be added to confirm this statement.

- **Answer: Thank you for your comment. We consent to your comment. So, we will add references about results in soil and hydrologic change by climate change in South Korea.**

6) The discussion Specific comments and technical corrections: Page 2, lines 4-5: I would suggest replacing "will" with "may", "might" or "could"

- **Answer: Thank you for your comment. We consent to your comment. So, we will correct this.**

7) The discussion Specific comments and technical corrections: Page 2, line18: Why is only Ethiopia mentioned here? These kinds of models were also used in other countries.

- **Answer: Thank you for your comment. We consent to your comment. So, we will add more literature from other countries.**

8) The discussion Specific comments and technical corrections: Page 2, line 33: This sentence should be rephrased because the TC equation is defined on the next page of the manuscript.

-   **Answer: Thank you for your comment. We consent to your comment. So, we will correct WATEM/SEDEM algorithm instead of TC equation.**

**9)** The discussion Specific comments and technical corrections: Page 3, line 8: Which spatially semi-distributed model?

-   **Answer: Thank you for your comment. We consent to your comment. As you know, WATEM/SEDEM is a spatially-distributed soil erosion and sediment transport model based on the RUSLE model plus a sediment transport capacity equation. We will removal semi distributed model and write spatially-distributed model. But, What we expressed the model as semi distributed model is because WATEM/SEDEM model used input R factor as single value. That is why we explain WATEM/SEDEM model as semi-distributed model.**

10) The discussion Specific comments and technical corrections: Page 3, line 9: Replace "TC equation given" with "TC equation is given".

-   **Answer: Thank you for your comment. We consent to your comment. So, we will correct this.**

11) The discussion Specific comments and technical corrections: Page 3, lines 14-15: This modification of the algorithm should be described.

-   **Answer: Thank you for your comment. Original algorithm did not use spatial KTC and R factor data. The factors only used single factors on WATEM/SEDEM model. So. original algorithm was modified for using spatially distributed KTC and R factor. And then, we developed automatic pre-processing algorithm about spatial input data for TC equation by Python code. Also, we developed TC equation by Python code for spatially calculating all the input data. We consent to your comment. So, we will correct this. We will describe the modification of algorithm in part of 3 Results and discussion.**

12) The discussion Specific comments and technical corrections: Page 4, line 21: Replace "stations locate" with "stations are located".

-   **Answer: Thank you for your comment. We consent to your comment. So, we will correct this.**

13) The discussion Specific comments and technical corrections: Page 5, first paragraph of

section 2.4: This paragraph should be rewritten because it is not clear.

- **Answer: Thank you for your comment. We consent to your comment. So, we will correct this. As we are mentioned at comment 1, model implantation is divided into 4 sectors. So, we will rewrite model implantation by dividing it into four section in detail.**

14) The discussion Specific comments and technical corrections: Page 5, lines 9-10: How does the SWAT model verify measured suspended solids? Please rephrase this sentence.

- **Answer: Thank you for your comment. We consent to your comment. The whole SWAT model calibration process were proceeded by LH-OAT and OAT (One Factor At a Time) methods. SWAT calibration methods are explained by sensitivity parameters and evaluation. In part of 3 Results and discussion, we will add contents of sensitivity analysis and calibration process. In this parts, we will explain model verification using OAT method about streamflow, evapotranspiration, soil moisture and suspended solid.**

15) The discussion Specific comments and technical corrections: Page 6, line 5: Replace "The yearly" with "the yearly".

- **Answer: Thank you for your comment. We consent to your comment. So, we will correct this.**

16) The discussion Specific comments and technical corrections: Page 6, line 5: It should be described how was the rainfall erosivity calculated (which equations were used, how was the data measured, was 1-minute data really used or was the data aggregated,...).

- **Answer: Thank you for your comment. We consent to your comment. This paper was not explained about generation of 1 minute data and rainfall erosivity, calibration of SWAT model. Because methods for estimating rainfall erosivity and SWAT calibration are generally known in hydrology field, we don't mention detailed process. Also, we suggested a previous study thesis instead of detailed explanation, for example, the Ahn and Kim (2016). According to reviewer comment, we can give all the process in detail. Therefore, we will certainly explain generation of rainfall erosivity, data used in this study, and method of SWAT calibration in part of 2 Materials and methods.**

17) The discussion Specific comments and technical corrections: Page 6, second paragraph of section 3.1: This part should also be rewritten because several things are not well explained, for example: what is meant by yearly distributed sediment delivery? When you are referring to sediment delivery, is this the sediment delivery ratio or something else because the sediment delivery ratio should not have any units? The TC parameter in the

WATEM/SEDEM is the transport capacity parameter of each grid cell? How did you take into account the fact that the WATEM/SEDEM model should be used with grid resolution of 20 by 20 m and using different resolutions may cause problems (e.g., LS factor)? How was the LS factor calculated? How did you determine the soil erodibility factor?

- **Answer: Thank you for your comment. We consent to your comment. So, we will correct this.**

- **The transport capacity parameter in TC equation consist of each grid cell (1km by 1km) because of other grid cell parameters. According to your comment, it might cause some problems. I thought what you consider LS factor experimental field of 22.13 m by Wischmier & Smith(1965). So, you might mention cell size of 20 by 20m. I think that this problem will be solved by suggesting detailed method and equation for estimation of LS factor.**

- **In (R)USLE equation, L is a slope length factor and S is a slope factor. This equation was tested based on standard slope such as slope length(22.13m) and slope grade(9%). The LS factor at this time defined 1.0 value. For application of other conditions, Wischmier & Smith(1965) proposed the equation which can find out LS factor by changes of relative length and slope compared as standard condition.**

- $$LS = (\gamma/22.13)^m \cdot (65.41sin^2 + 4.56sin\theta + 0.065)$$
  **Where, $\gamma$ = slope length, $\theta$ = slope grade, m = slope index**

- **We consent to your comment. So, we used this equation. we will add LS factor method and results in detail as paragraph.**

- **Also, we will correct yearly distributed sediment delivery and sediment delivery ratio.**

18) The discussion Specific comments and technical corrections: Page 6, line 5: It should be described how was the rainfall erosivity calculated (which equations were used, how was the data measured, was 1-minute data really used or was the data aggregated,...).

- **Answer: We consent to your comments. This paper was not explained about generation of 1 minute data and rainfall erosivity. Because methods for estimating rainfall erosivity is generally known in hydrology field, we don't mention detailed process. According to reviewer comment, we can give all the process in detail. Therefore, we will certainly explain generation of rainfall erosivity, data used in this study in part of 2 Materials and methods. Also, we will suggest a previous study literature.**

19) The discussion Specific comments and technical corrections: Page 8, line 2: How was the calibration carried out? Page 8, lines 6-7: NSE and PBIAS acronyms should be defined.

- **Answer: Thank you for your comment. We consent to your comment. This paper was not explained about calibration of SWAT model. Because methods for**

estimating SWAT calibration are generally known in hydrology field, we don't mention detailed process. The whole SWAT model calibration process were proceeded by LH-OAT and OAT (One Factor At a Time) methods. SWAT calibration methods are explained by sensitivity parameters and evaluation. In part of 3 Results and discussion, we will add contents of sensitivity analysis and calibration process. In this parts, we will explain model verification using OAT method about streamflow, evapotranspiration, soil moisture and suspended solid.

- **We will add definition and application of NSE and PBIAS. Also, we will suggest a detailed previous literature about NSE and PBIAS by moriasi(2007).**

20) The discussion Specific comments and technical corrections: Page 8, lines 10-11: More information should be provided about the soil moisture data. It is stated that detailed results are available in the paper that is currently under review, but this paper is publically not available.

- **Answer: We consent to your comments. We used soil moisture data at observed flux data by KICT (Korea Institute of Civil engineering and building Technology). So, we will correct this and add sentences in part of 2 Materials and methods.**
- **Also, paper by Ahn and Kim (2016) was finally accepted. So, we will correct this.**

21) The discussion Specific comments and technical corrections: Page 8, line 15: What is meant by "eight days intervals"? Please explain. Page 8, line 18: "indicates a satisfactory simulation". This is very subjective and I would suggest avoiding this kind of statements.

- **Answer: We consent to your comments. The eight days interval mean measured interval or cycle. Sediment data is not daily measured data. Therefore, It is measured every eight day. So, we will correct this and add sentences in part of 2 Materials and methods.**
- **Also, we used $R^2$, NSE and PBIAS index for assessment of model results. We consent to your comments about subjective satisfactory of model results. The index evaluation of $R^2$ are NSE are generally known in hydrology field. Donigian (2000) reported that a $R^2$ for daily flow less than 0.6, from 0.6 to 0.7, from 0.7 to 0.8, and greater than 0.8 are classified as poor, fair, good, and very good, respectively. Moriasi et al. (2007) recommended the performance ratings of NSE for monthly flows less than 0.5, from 0.5 to 0.65, from 0.65 to 0.75, and greater than 0.75 are unsatisfactory, satisfactory, good, and very good, respectively. We will add definition and application of NSE and PBIAS. Also, we will suggest a detailed previous literature about NSE and PBIAS by moriasi(2007).**

22) The discussion Specific comments and technical corrections: Page 9, Fig. 5: I would suggest using different colours for observed and simulated data(one could be red and the other one grey or black; or something similar).

- **Answer: Thank you for your comment. We consent to your comment. So, we will correct this.**

[Figure]

23) The discussion Specific comments and technical corrections: Page 9, Page 10, Fig. 6: I
would suggest using different colours for observed and simulated data (one could be red
and the other one grey or black; or something similar).

- **Answer: Thank you for your comment. We consent to your comment. So, we will correct this.**

[Figure]

24) The discussion Specific comments and technical corrections: Page 11, Fig. 7: Are these really daily suspended solids because the density of points is a bit low (e.g., c) case). How were suspended solids measured and how reliable are measurements?

- **Answer: Thank you for your comment. We consent to your comment. As I was mentioned, the eight days interval mean measured interval or cycle. Suspended solid data is not daily measured data. Therefore, It is measured every eight day.**
- **Suspended solid data have be collected from the monitoring network operated by Ministry of Environment in South Korea. By law article 22 of environmental policy and water quality monitoring network, these data actually use basic data for analyzing the effect of environmental policy and establishing policy at the national level in South Korea. Also, this data are verified to two levels by NIER (National Institute of Environmental Research). NIER classify of measured Suspended solid outlier by statistical analysis and continually improve the problem(Ministry of Environment, 2016). So, suspended solid data which we used verified reliability up to 90%. We consent to your comment. We will add a sentence above and suggest a detailed literature about Suspended solid reliability by Ministry of Environment(2016).**
- **Ministry of Environment (2016). "Water quality monitoring program".**

25) The discussion Specific comments and technical corrections: Page 12, first paragraph of section 3.3: This section should be better explained.

- **Answer: Thank you for your comment. We consent to your comment. So, we will correct this.**

26) The discussion Specific comments and technical corrections: Page 14, lines 2-3: Why is it difficult to estimate the KTC and what is the uncertainty related to this estimation?

- **Answer: Thank you for your comment. We consent to your comment. As KTC is empirical coefficient, KTC was estimated by comparing observed sediment at small experimental field in previous study or uniformly used as 1.0 or 100 value at overall watershed. But, measurement of sediment in all areas is impossible. Therefore, we think that KTC need a method for estimation depending on different land uses and watershed characteristics at small watershed. We will add a sentence above.**
- **Also, we think that uncertainty related to this estimation depend on calibration result of SWAT model. So, We will add a sentence above.**

27) The discussion Specific comments and technical corrections: Page 14, lines 3-4: Please rephrase this sentence. Page 14, lines 9-10: Please rephrase this sentence, because it is not clear.

- **Answer: Thank you for your comment. We consent to your comment. So, we will correct this.**

28) The discussion Specific comments and technical corrections: Page 15, lines 15-16: This is subjective (what is high and what is low?). Page 15, lines 15-16: This is subjective (what is high and what is low?).

- **Answer: Thank you for your comment. We consent to your comment. We used $R^2$ for assessment of KTC equation. We consent to your comments about subjective satisfactory of model results. The index evaluation of $R^2$ are NSE are generally known in hydrology field. Donigian (2000) reported that a $R^2$ for daily flow less than 0.6, from 0.6 to 0.7, from 0.7 to 0.8, and greater than 0.8 are classified as poor, fair, good, and very good, respectively. Moriasi et al. (2007) recommended the performance ratings of NSE for monthly flows less than 0.5, from 0.5 to 0.65, from 0.65 to 0.75, and greater than 0.75 are unsatisfactory, satisfactory, good, and very good, respectively. We will add definition and application of NSE and PBIAS. Also, we will suggest a detailed previous literature about NSE and PBIAS by moriasi(2007).**

29) The discussion Specific comments and technical corrections: Page 17, section 3.6: This section should be moved before the results and discussion part. From my points of view this should go into the methodology. More information about the software could be provided (e.g., is it publically available,. . .).

- **Answer: Thank you for your comment. We consent to your comment. So, we will correct this. We will add input/output algorithm regarding to rainfall erosivity program and TC modeling program respectively.**

30) The discussion Specific comments and technical corrections: Page 17, line 17: "The KTC was traced" this is not clear, please rephrase.

- **Answer: Thank you for your comment. We consent to your comment. So, we will correct this and add figure for trace process below.**

[Figure]

31) The discussion Specific comments and technical corrections: Page 20, lines 7-8: This reference is not mentioned in text.

- **Answer: Thank you for your comment. We consent to your comment. So, we will remove this reference.**

---

## Author Comment (AC2) · 6 Apr 2017

The comment was uploaded in the form of a supplement: http://www.hydrol-earth-syst-sci-discuss.net/hess-2016-649/hess-2016-649-AC2- supplement.pdf

HESS-2016-649-RC2 We thank the reviewer for his constructive review and intend to address all of his comments. 1) The paper structure is a bit confused, the main objective of the study or the scientific question is not clear enough. Further, what are

actually the main conclusions of the study, what is the take home message of this paper? Moreover, the language should be significantly improved (grammar, overall style and structure because some sentences are not clear). In its current form the paper is not suitable for the Hydrology and Earth System Sciences journal. The submission describes the estimation of transport capacity coefficient (KTC) in WATEM/SEDEM algorithm with the evaluation of RUSLE R factor using 1 min rainfall data in Han River basin of South Korea. The SWAT model, which includes the MUSLE function for calculating soil losses from the watershed, has been used to determine the WATEM/SEDEM sediment transport estimation. Studies such as this are relatively rare, and the model appears to be effectively calibrated and applied. This reviewer agrees that the manuscript contains novel information that could be useful for the readers of HESS. Much of the theoretical development presented in this manuscript is clear and well described. However, it reads more like a book chapter than a journal article. It is because the authors present few theoretical background and discussion of results, implications and limitations. For example, there is a lack of information regarding how the variation of KTC could affect the sediment yield at the sub-watershed scale. Moreover, several sections of the manuscript are not connected well, and importantly it is hard to understand what the major findings are. Although I generally recommend the paper for publication in HESS, I have the following comments which have, to my opinion, to be considered in a revised version. - Answer: Thank you for your comment. We consent to your comment. We think that all the following comments will go through from all your comments.

2) Introduction. P2.L29-32: it confuses me why you used such long content to introduce SWAT studies, which are not the key topic of your study. I would like see a clear hypothesis (framework) of your study, following introduction of your aim line P2.L33-P3.L3. Then, if essentially, introduce some method to test your hypothesis. - Answer: Thank you for your comment. We consent to your comments. As you know, main objective of this paper is to fully develop distributed WATEM/SEDEM algorithm and assess KTC equation for estimation of KTC factor. At here, SWAT simulated daily sediment was assumed as daily observed sediment by calibration with measure 8-

days sediment. Therefore, SWAT results serve as input data for estimation of KTC equation. So, we will correct this. We will remove sentences regarding SWAT studies and add paragraphs regarding introduce some method to test your hypothesis in part of 2 Materials and methods. 3) Study area description. P4.L13-21: please introduce rough annual distribution of precipitation and temperature, e.g. precipitation mostly occurred in some month, min and max temperature over year. Add a description of land use and soil data modeled in this study. How were point sources of sediment, N and P accounted for? Figure 1: Please remove the layers that were not used in model calibration. - Answer: Thank you for your comment. We consent to your comment. Han River Basin in South Korea is either classified as a humid subtropical climate. Summers are generally hot and humid, with the East Asian monsoon taking place from June until September. August, the warmest month, has average high and low temperatures of 29.6 and 22.4 °C with higher temperatures possible. Winters are often cold to freezing with average January high and low temperatures of 1.5 and -5.9 °C and are generally much drier than summers, with an average of 28 days of snow annually. Sometimes, temperatures do drop dramatically to below -10.0 °C, in odd occasions rarely as low as -15.0 °C in the mid-winter period between January and February. An average slope of 35.9% and an average elevation of 404.7 m. More than 73.3% (25,030 km2) of the watershed area is forested, and 12.2% (811 km2) is cultivated. The cultivated area consists of 1,699 km2 of paddy fields and 3,554 km2 of upland crops. The dominant soil is sandy loam (51.0%). So, we will add watershed description above. - Also, point sources mean sewage discharge. At point sources shown Figure 1, Domestic, agricultural, industrial water are treated and discharge such as flow, sediment, nitrogen, and phosphorus. The application of point sources in SWAT model improves accuracy of watershed modeling. So, application of point sources is required. We used point source data from the Ministry of Environment in South Korea. So, we will add description of point source data above.

4) Method: Authors should provide proper justification to consider this approach for possible use in other studies. The differences and limitations should be included in

the Methodology. - Answer: Thank you for your comment. We consent to your comment. We will certainly explain differences and limitations compared as other studies. We would like to state that the presented paper included various study such as models, algorithm and regression analysis. In this paper, TC equation in WATEM/SEDEM algorithm was firstly introduced to South Korea. The one of limitations is that WATEM/SEDEM algorithm can't consider land use compared as RUSLE. Someone can recommend the RUSLE equation than this algorithm. In order to improve the problem, we additionally regenerated KTC by considering agricultural area. In TC equation, we think that characteristics of land use represent KTC ranges. So, KTC shows difference between forest and agriculture. We will add differences and limitations above in the Methodology.

5) Model implementation. P5.L4-15: More detail about soils how similar were the attributes (e.g. soil type) of the sub-watersheds. How were data for the individual KTC determined? - Answer: Thank you for your comment. We consent to your comment. For estimation of KTC, K factor based on RUSLE equation is used as input data. By generating soil texture, K factor is estimated. As you ask, soil distribution is very important in estimating KTC factors. We didn't explain soil and K factor distribution. So, we will describe soil and K factor distribution for checking attribute of soils in sub-watersheds at revised manuscript.

6) Results and discussion: Overall, the authors failed to provide a detailed report on the data obtained during the study and then need to discuss the importance of this study with regard to the relevant scientific or technical issues about sediment transport capacity. In this section, the authors simply explained the outcomes from model simulation that could not support to the significant results. Discussion should be concise and add only essential points in terms of the current results and limitations. - Answer: Thank you for your comment. We consent to your comment. This paper was not explained about generation of major data (rainfall erosivity, 1minute rainfall data, suspended solid, soil moisture, K factor, Soil. . .) for this study. We used soil moisture

data at observed flux data by KICT (Korea Institute of Civil engineering and building Technology). Overall, we essentially didn't describe data sources and method for generation in detail. We will correct this and add sentences in part of 2 Materials and methods. - Also, we agree with your discussion frame. The essential point of this study is to estimate sediment transport capacity and to KTC empirical equation for sediment transport capacity from results of SWAT and TC equation. Therefore, in order to apply accurate TC equation in South Korea, KTC value from KTC empirical equation is essential. So, we will rewrite three important points in Results and discussion. 1st point is a summary of SWAT and TC model results, 2nd point is to describe current results and limitations, 3rd is review the causes of uncertainty about KTC empirical equation.

7) Conclusions: The findings of this study will be more useful if the authors can address how these findings will impact the evaluation of sediment transport capacity. Conclusions could be better stated by a better interpretation of the data and model predictions. - Answer: Thank you for your comment. We consent to your comment. There are two final findings in this paper. 1st finding is implementation of TC modeling for sediment transport capacity in South Korea and 2nd finding is estimation of KTC empirical equation. These findings could use more easily soil transport modeling compared to RUSLE and MUSLE because of simple input data. Also, using suitable empirical equation for South Korea, it is possible to predict the correct results. Therefore, the modeling proposed in this study could be recommended for soil transport or soil yield in ungauged watershed and areas in South Korea, because South Korea is mostly mountainous and difficult to measure data.

Please also note the supplement to this comment:
http://www.hydrol-earth-syst-sci-discuss.net/hess-2016-649/hess-2016-649-AC2-supplement.pdf
* * *

---

## Referee Comment (RC3) · Anonymous Referee #3 · 12 Apr 2017

While the research focus is in line with the journal scope and aims, the paper in its current form is far from being ready for publication. The paper needs a thorough English review. As of now the way it is written makes it very complicated to understand the research itself. Further major concerns are related to the paper structure, which appears very disorganised, in addition to the paper lack of clarity. As well, the material and methods are not sufficiently described, thus making this study unreproducible to others. The authors describe the two main sediment delivery method and SWAT

model that are already coded and well known in the literature. However, they leave out a huge part of their method. How was the rainfall erosivity determined based on the 1-minute rainfall data? The authors do not describe the method used to define a spatial distribution of the rainfall erosivity, yet they presented it in the discussion session. The methods do not describe how they calibrated their models. Many info on the study site, pertaining the method itself, are also missing (measured sediment? Soil characteristics?) Finally, the discussion itself is not broad enough to be convincing about the research value. Few lines are spent to describe the results in each subchapter; the reader is left just to see nice figures, that are however barely described in the text. I do not feel this is enough for a scientific paper. As well part of the results are actually methods.

Some more detailed comments follow ABSTRACT Abstract needs rephrasing. Aside rom the english form, as it is now, it is very dense in acronyms and numbers, and this makes it hard to read. There is too much focus on the methods that, however, still do not appear clear to the reader due to the sentences structures. Furthermore, methods are mixed with results as well. The point of the abstract should be to be as clear as possible to give the idea of the study: as of now this is not accomplished at all. What is the aim of the study? The authors state is to estimate watershed scale sediment yield distribution, but the number given in the abstract refers to the rain erosivity, spatial KTC, and about the sediment yield the only information given is the relationship with the measured values. The paper afterwords describe a different aim of the study, which is to evaluate the KTC...

INTRODUCTION The introduction is very redundant and doesn't really get the point across. The whole introduction should be rephrased and reorganised. Line 9 to 16 page 2. This whole part is a mess. Line 10-11 do not make sense: "However, spatial data is often scarce possibilities to model spatial patterns of sediment delivery and to identify source areas of sediment are very limited". While this is a citation from another work, I think the author are missing some verbs or words or periods... Line 15-16 page

2: "the sediment delivery ratio needs to be determine to generate the sediment."???
What does this mean? The sediment delivery ratio does not generate sediment... Line
17-20 page 2: numerous models exist. As the text is written now it seems that these
models have been applied only in Ethiopia. I do not think this is true. If the authors
choose to speak about each model and the locations where it has been applied, they
should be consistent and nominate all the locations for all models. If not, the fact that
it has been applied in Ethiopia can be removed. Line 27-28 page 2: you can put the
references together, no need to describe the specific application to Spain. Line 2 and 3
of page 3 states "the KTC is traced by the sediment delivery of SWAT model determined
by comparing the MUSLE based SWAT simulated sediment yield", this sentence is very
unclear. What is the aim of the study? The authors in the abstract state they want to
evaluate the watershed-scale sediment yield distribution, but in the introduction, they
state they want to assess the KTC (Transport Capacity Coefficient) of the TC equation,
so which one is it? The abstract speaks of 14 years of rain; the introduction speaks
about 15 years, please be consistent.

METHODS The authors describe the two main sediment delivery method and SWAT
model, that are already coded and well known in the literature. However, they leave
out a huge part of their method. How was the rainfall erosivity determined based on
the 1-minute rainfall data? The authors do not describe the method used to define a
spatial distribution of the rainfall erosivity, nor how they calibrated the model. Many info
on the study site, pertaining the method itself, are also missing (measured sediment?
Soil characteristics?)

DISCUSSIONS The discussion is very short. Too much. There is no mention of any
relationship with other studies in the literature, nor details on how these results could
be useful for other researchers in the field. The discussion is mainly spent in figures,
which are however barely described or commented. just to mention some issues, in
chapter 3.1 the maps are not described in the text, are they a result? Are they part
of the method? Where do they come from? What is their use? In chapter 3.2 the

authors seem to focus more on calibration and validation, which is part of the method. However, they do not give enough information, and they refer the reader to unpublished studies (e.g. line 12 page 8)...

Overall I think this paper needs a deep rethinking and reorganization before being ready for publication.

---

## Author Comment (AC3) · 2 May 2017

You can see the supplement file.

HESS-2016-649-RC3 We thank the reviewer for his constructive review and intend to address all of his comments. We would like to state that the presented paper included various study such as models, algorithm and regression analysis. In this paper, WATEM/SEDEM algorithm was firstly introduced to South Korea. Also, we think that KTC

empirical equation would be useful at ungauged watershed. For resubmission of improved paper, we think that all the following comments went through from the your reviews.

1) The paper needs a thorough English review. As of now the way it is written makes it very complicated to understand the research itself.

- Answer: Thank you for your comment. We consent to your comments. For language problems, we plan to get the native English speaker review/proofread through American Journal Experts (payment is about $ 500). We will attach editing invoice.

2) Further major concerns are related to the paper structure, which appears very disorganised, in addition to the paper lack of clarity. As well, the material and methods are not sufficiently described, thus making this study unreproducible to others.

- Answer: The paper is outlined as follows: Sect. 1 described application of WATEM/SEDEM algorithm in South Korea. However, KTC (Transport Capacity Coefficient) is necessary for application of WATEM/SEDEM algorithm in South Korea. So, Sect. 2 traced KTC by the sediment delivery of SWAT model determined as comparing MUSLE (Modified USLE) based SWAT (Soil Water Assessment Tool) simulated sediment yield. The SWAT model results reflected observed suspended solid. Sect. 3 find out KTC empirical equation by linear regression analysis. The KTC equation is going to be commonly used for accurate sediment delivery at a ungauged watershed in South Korea. Finally, calibrated spatial sediment delivery from WATEM/SEDEM algorithm is estimated using obtained KTCs by KTC empirical equation. - The objective of this study is to estimate KTC empirical equation for calibrated spatial sediment delivery and to prove accuracy of sediment delivery by KTC empirical equation. Therefore, we consent to reviewer's comments. We are going to rewrite paper structure according to above purpose.

3) The authors describe the two main sediment delivery method and SWAT model that are already coded and well known in the literature. However, they leave out a

huge part of their method. How was the rainfall erosivity determined based on the 1-minute rainfall data? The authors do not describe the method used to define a spatial distribution of the rainfall erosivity, yet they presented it in the discussion session. The methods do not describe how they calibrated their models.

- Answer: Thank you for your comment. We consent to your comments. This paper was not explained about generation of 1 minute data and rainfall erosivity, calibration of SWAT model. Because methods for estimating rainfall erosivity and SWAT calibration are generally known in hydrology field, we don't mention detailed process. Also, we suggested a previous study thesis instead of detailed explanation, for example, the Ahn and Kim (2016). According to reviewer comment, we can give all the process in detail. Therefore, we will certainly explain generation of rainfall erosivity, data used in this study, and method of SWAT calibration in part of 2 Materials and methods.

4) Many info on the study site, pertaining the method itself, are also missing (measured sediment? Soil characteristics?) Finally, the discussion itself is not broad enough to be convincing about the research value. Few lines are spent to describe the results in each subchapter; the reader is left just to see nice figures, that are however barely described in the text. I do not feel this is enough for a scientific paper. As well part of the results are actually.

- Answer: Thank you for your comment. We consent to your comment. By reflecting your comments, discussion about results will be improved. Also, we will additionally analyze cause of error according to each sector. So, we will make up for the weak points in this paper. We will certainly explain soil characteristics in regard to rill erosion factor. We used soil moisture data at observed flux data by KICT (Korea Institute of Civil engineering and building Technology). Overall, we essentially didn't describe data sources and method for generation in detail. We will correct this and add sentences in part of 2 Materials and methods. - Answer: This paper was not explained about generation of major data (rainfall erosivity, 1 minute rainfall data, suspended solid, soil moisture, K factor, Soil…) for this study. For the reasons, we thought that the discussion has not sufficiently explained. Most of all, we must address how these findings will impact the evaluation of sediment transport capacity final purpose of this study. - There are two final findings in this paper. 1st finding is implementation of TC modeling for sediment transport capacity in South Korea and 2nd finding is estimation of KTC empirical equation. These findings could use more easily soil transport modeling compared to RUSLE and MUSLE because of simple input data. Also, using suitable empirical equation for South Korea, it is possible to predict the correct results. Therefore, the modeling proposed in this study could be recommended for soil transport or soil yield in ungauged watershed and areas in South Korea, because South Korea is mostly mountainous and difficult to measure data.

5) Some more detailed comments follow ABSTRACT Abstract needs rephrasing. Aside rom the english form, as it is now, it is very dense in acronyms and numbers, and this makes it hard to read. There is too much focus on the methods that, however, still do not appear clear to the reader due to the sentences structures. Furthermore, methods are mixed with results as well. The point of the abstract should be to be as clear as possible to give the idea of the study: as of now this is not accomplished at all. What is the aim of the study? The authors state is to estimate watershed scale sediment yield distribution, but the number given in the abstract refers to the rain erosivity, spatial KTC, and about the sediment yield the only information given is the relationship with the measured values. The paper afterwords describe a different aim of the study, which is to evaluate the KTC....

- Answer: This paper was not explained about generation of major data (rainfall erosivity, 1 minute rainfall data, suspended solid, soil moisture, K factor, Soil...) for this study. For the reasons, we thought that the discussion has not sufficiently explained. Most of all, we must address how these findings will impact the evaluation of sediment transport capacity final purpose of this study. - There are two final findings in this paper. 1st finding is implementation of TC modeling for sediment transport capacity in South Korea and 2nd finding is estimation of KTC empirical equation. These findings could

use more easily soil transport modeling compared to RUSLE and MUSLE because of simple input data. Also, using suitable empirical equation for South Korea, it is possible to predict the correct results. Therefore, the modeling proposed in this study could be recommended for soil transport or soil yield in ungauged watershed and areas in South Korea, because South Korea is mostly mountainous and difficult to measure data.

6) INTRODUCTION The introduction is very redundant and doesn't really get the point across. The whole introduction should be rephrased and reorganised. Line 9 to 16 page 2. This whole part is a mess. Line 10-11 do not make sense: "However, spatial data is often scarce possibilities to model spatial patterns of sediment delivery and to identify source areas of sediment are very limited". While this is a citation from another work, I think the author are missing some verbs or words or periods...

- Answer: We wanted to mention importance of spatial pattern and data in regard to soil erosion. On your comments, the introduction wasn't clear. We will rewrite these paragraphs to improve completeness below. "The soil erosion and sedimentation require a basic understanding about the spatial processes of transport capacity. For understanding spatial process, spatial topography data are essentially needed. However, acquisition and generation of spatial data are very difficult (Haregeweyn et al., 2013). The soil erosion is related to the water flow and topography factors which is controlled by the abundance and type of vegetation, underlying soil, elevation."

7) Line 15-16 page? 2: "the sediment delivery ratio needs to be determine to generate the sediment."??? What does this mean? The sediment delivery ratio does not generate sediment. . . Line 17-20 page 2: numerous models exist. As the text is written now it seems that these models have been applied only in Ethiopia. I do not think this is true. If the authors choose to speak about each model and the locations where it has been applied, they should be consistent and nominate all the locations for all models. If not, the fact that it has been applied in Ethiopia can be removed.

- Answer: Thank you for your comment. We consent to your comment. These models

are not the models which were only applied in Ethiopia. The ANSP, LSEM, and SWAT models are widely used throughout the world for soil erosion. Therefore, we will suggest a previous study thesis about application of models and remove word or Ethiopia according to reviewer comment.

8) Line 27-28 page 2: you can put the references together, no need to describe the specific application to Spain.

- Answer: Thank you for your comment. We consent to your comment. The sentence is not about the only model application. The model is widely used throughout the world for soil erosion. We will describe various model application.

9) Line 2 and 3 of page 3 states "the KTC is traced by the sediment delivery of SWAT model determined by comparing the MUSLE based SWAT simulated sediment yield", this sentence is very unclear. What is the aim of the study? The authors in the abstract state they want to evaluate the watershed-scale sediment yield distribution, but in the introduction, they state they want to assess the KTC (Transport Capacity Coefficient) of the TC equation, so which one is it? The abstract speaks of 14 years of rain; the introduction speaks about 15 years, please be consistent.

- Answer: Thank you for your comment. We consent to your comment. We will consistently correct study period. The paper is outlined as follows: Sect. 1 described application of WATEM/SEDEM algorithm in South Korea. However, KTC (Transport Capacity Coefficient) is necessary for application of WATEM/SEDEM algorithm in South Korea. So, Sect. 2 traced KTC by the sediment delivery of SWAT model determined as comparing MUSLE (Modified USLE) based SWAT (Soil Water Assessment Tool) simulated sediment yield. The SWAT model results reflected observed suspended solid. Sect. 3 find out KTC empirical equation by linear regression analysis. The KTC equation is going to be commonly used for accurate sediment delivery at a ungauged watershed in South Korea. Finally, calibrated spatial sediment delivery from WATEM/SEDEM algorithm is estimated using obtained KTCs by KTC empirical equation. - The objective of

this study is to estimate KTC empirical equation for calibrated spatial sediment delivery and to prove accuracy of sediment delivery by KTC empirical equation. Therefore, we consent to reviewer's comments. We are going to rewrite paper structure according to above purpose.

10) METHODS The authors describe the two main sediment delivery method and SWAT model, that are already coded and well known in the literature. However, they leave out a huge part of their method. How was the rainfall erosivity determined based on the 1-minute rainfall data? The authors do not describe the method used to define a spatial distribution of the rainfall erosivity, nor how they calibrated the model. Many info on the study site, pertaining the method itself, are also missing (measured sediment? Soil characteristics?).

- Answer: Thank you for your comment. We consent to your comment. This paper was not explained about generation of major data (rainfall erosivity, 1minute rainfall data, suspended solid, soil moisture, K factor, Soil...) for this study. We used soil moisture data at observed flux data by KICT (Korea Institute of Civil engineering and building Technology). Overall, we essentially didn't describe data sources and method for generation in detail. We will correct this and add sentences in part of 2 Materials and methods. - Also, we agree with your discussion frame. The essential point of this study is to estimate sediment transport capacity and to KTC empirical equation for sediment transport capacity from results of SWAT and TC equation. Therefore, in order to apply accurate TC equation in South Korea, KTC value from KTC empirical equation is essential. So, we will rewrite three important points in Results and discussion. 1st point is a summary of SWAT and TC model results, 2nd point is to describe current results and limitations, 3rd is review the causes of uncertainty about KTC empirical equation.

11) DISCUSSIONS The discussion is very short. Too much. There is no mention of any relationship with other studies in the literature, nor details on how these results could be useful for other researchers in the field. The discussion is mainly spent in figures,

which are however barely described or commented. just to mention some issues, in chapter 3.1 the maps are not described in the text, are they a result? Are they part of the method? Where do they come from? What is their use? In chapter 3.2 the authors seem to focus more on calibration and validation, which is part of the method. However, they do not give enough information, and they refer the reader to unpublished studies (e.g. line 12 page 8)....

- Answer: Thank you for your comment. We consent to your comment. As you mentioned, this paper was not explained about generation of major data (rainfall erosivity, 1 minute rainfall data, suspended solid, soil moisture, K factor, Soil...) and detailed process for this study. For the reasons, we thought that the discussion has not sufficiently explained. Most of all, we must address how these findings will impact the evaluation of sediment transport capacity final purpose of this study. Also, we will compare final founding with other study results. However, there were no comparable domestic studies, because TC equation in WATEM/SEDEM algorithm was firstly introduced to South Korea in this paper. Finally, we are believed that development of KTC equation can be used as a tool to improve the results of existing overseas research. This is the conclusion in this study. On your comment. The discussion about results will be improved. - Also, paper by Ahn and Kim (2016) was finally accepted. So, we will correct this.

Please also note the supplement to this comment:
http://www.hydrol-earth-syst-sci-discuss.net/hess-2016-649/hess-2016-649-AC3-supplement.pdf